# Impact of Clinical Use of Probiotics on Preterm-Related Outcomes in Infants with Extremely Low Birth Weight

**DOI:** 10.3390/nu16172995

**Published:** 2024-09-05

**Authors:** Wei-Hung Wu, Ming-Chou Chiang, Ren-Huei Fu, Mei-Yin Lai, I-Hsyuan Wu, Reyin Lien, Chien-Chung Lee

**Affiliations:** Division of Neonatology, Department of Pediatrics, Chang Gung Memorial Hospital, School of Medicine, Chang Gung University, Taoyuan 333, Taiwan; ch60819@cgmh.org.tw (W.-H.W.); cmc123@cgmh.org.tw (M.-C.C.); rkenny@cgmh.org.tw (R.-H.F.); a9275@cgmh.org.tw (M.-Y.L.); a9277@cgmh.org.tw (I.-H.W.); reyinl@cgmh.org.tw (R.L.)

**Keywords:** extremely low birth weight, mortality, probiotics

## Abstract

**Background:** Preterm birth significantly contributes to mortality and morbidities, with recent studies linking these issues to gut microbiota imbalances. Probiotic supplementation shows promise in mitigating adverse outcomes in preterm infants, but optimal timing and guidelines remain unclear. This study assesses the benefits of probiotic supplementation for preterm infants without consistent guidelines. **Methods:** This retrospective study examined extremely low-birth-weight (ELBW) infants in neonatal intensive care units from 2017 to 2021. Mortality and preterm-related outcomes were compared between infants receiving probiotics and those not. Subgroup analyses based on probiotic initiation timing were conducted: early (≤14 days), late (>14 days), and non-probiotic groups. **Results:** The study included 330 ELBW infants: 206 received probiotics (60 early, 146 late), while 124 did not. Probiotic supplementation was associated with lower overall mortality (adjusted OR 0.22, 95% CI 0.09–0.48) and decreased mortality from necrotizing enterocolitis (NEC) or late-onset sepsis (LOS) (adjusted OR 0.12, 95% CI 0.03–0.45). Early probiotics reduced overall mortality, NEC/LOS-related mortality, and NEC/LOS-unrelated mortality. Late probiotics decreased overall mortality and NEC/LOS-related mortality. Early probiotic use also expedited full enteral feeding achievement. **Conclusions:** Probiotic supplementation reduces mortality and improves feeding tolerance in preterm infants. Establishing guidelines for probiotic use in this population is crucial.

## 1. Introduction

Preterm birth, defined as a gestational age (GA) of less than 37 weeks, is one of the leading causes of mortality in children under the age of five [1,2]. Beyond its association with mortality, preterm birth is linked to a multitude of morbidities, including respiratory distress syndrome (RDS), intraventricular hemorrhage (IVH), bronchopulmonary dysplasia (BPD), necrotizing enterocolitis (NEC), retinopathy of prematurity (ROP), and periventricular leukomalacia (PVL) [3]. Recent studies utilizing next-generation sequencing suggest associations between these preterm morbidities and the gut microbiota, particularly in cases of NEC and sepsis [4]. 

The gut microbiota play a crucial role not only as a barrier against pathogenic bacteria and toxins but also in digestion and absorption [5]. Premature infants are particularly vulnerable to dysbiosis due not only to their premature birth but also to prolonged antibiotic use, mechanical ventilator support, and environmental factors. Early microbiome detection in the gut of preterm infants reveals initial colonization by pathogenic and hospital-associated microbiomes, including *Staphylococci*, *Enterococcus* spp., and *Klebsiella* spp. In contrast, term infants predominantly harbor microbiomes of vaginal, maternal fecal, or skin origin, dominated by *Lactobacilli* spp. or *Bifidobacterium* spp. Consequently, probiotic supplementation has been used for years in neonatal care to prevent morbidity and mortality, particularly in NEC and sepsis [6,7,8,9].

In recent years, probiotics have been demonstrated to be effective in preventing NEC, late-onset sepsis, and reducing mortality rates in preterm infants [10]. However, most of this evidence is derived from well-designed case–control randomized trials or experimental observational studies [11]. The practical efficacy of probiotics, particularly in managing multiple complex conditions and in the absence of established guidelines, remains uncertain. Additionally, the optimal timing for initiating probiotic supplementation remains unclear [9]. In our hospital, standardized guidelines for probiotic supplementation in preterm infants are still lacking, and probiotic use primarily relies on clinician judgment. Therefore, this study aims to investigate whether premature infants still derive benefits from probiotic supplementation based on individual clinician judgment in the absence of consistent guidelines. Additionally, we aim to explore whether the timing of probiotic initiation impacts its efficacy.

## 2. Materials and Methods

### 2.1. Study Participants

We conducted a retrospective review of extremely low-birth-weight (ELBW) infants admitted to the Neonatal Intensive Care Unit (NICU) at Lin-Kou Chang Gung Memorial Hospital, a tertiary-level medical center in northern Taiwan that handles approximately 800 deliveries annually. The review period spanned from January 2017 to December 2021. Infants with congenital gastrointestinal (GI) tract anomalies, spontaneous intestinal perforation (SIP), those who did not initiate feeding within the first two weeks of life, or those transferred to other hospitals were excluded from the study. The study participants were categorized into two groups: the probiotic group and the non-probiotic group, based on whether they received probiotics. The probiotic group was further subdivided into early probiotic group (probiotic initiation ≤ 14 days old) and late probiotic group (probiotic initiation > 14 days old) for subgroup analysis. In our NICU, we administered a daily dose of probiotics in conjunction with human milk or preterm formula. The choice of probiotic regimen is MoProbi-LR (150 mg/cap, containing *Lactobacillus rhamnosus GG* 10^9^ cfu/g) or Infloran (250 mg/cap, containing *Lactobacillus acidophilus* 150 × 10^9^ cfu/g and *Bifidobacterium bifidum* 150 × 10^9^ cfu/g) for a single dose, which is typically administered once a day via oral gastric tube diluted with milk. The nutritional table is shown in Appendix A. The choice of regimen and the timing of introduction are determined by the responsible neonatologists.

### 2.2. Neonatal Characteristic and Outcomes Measures

We collected data on GA, birth body weight (BBW), 1 and 5 min Apgar scores, small-for-gestational-age (SGA) status, grade of intraventricular hemorrhage (IVH), early-onset sepsis (EOS), and hemodynamically significant patent ductus arteriosus (HsPDA), including whether ligation was performed. We also documented the initiation time for enteral feeding and the time taken to reach complete enteral feeding (CEF). We considered mortality- and preterm-related outcomes such as NEC, late-onset sepsis (LOS), BPD, PVL, and ROP as the outcomes for comparison. 

EOS and LOS were determined based on the timing of onset and microbiological results from blood or CSF, with EOS developing within the first 3 days of life and LOS occurring after 3 days of life. The NEC definition used for analysis was modified Bell’s criteria ≥ stage 2. We included mortality from any cause, with subgroups for sepsis- and NEC-related mortality. Mortality associated with NEC and sepsis was defined as an event occurring within 7 days of the diagnosis. The definition of moderate BPD, classified as grade II or above, was based on the 2019 Jensen definition [12]. ROP was compared at ≥stage 2 with plus disease or ≥stage III without plus disease. The time to reach CEF was defined as the point at which total enteral feeding reached 120 mL/kg/day or total parenteral nutrition supplementation was discontinued”.

### 2.3. Statical Analysis

The normality of continuous parameters was assessed using the Kolmogorov–Smirnov test. Since these variables were not normally distributed, they are presented as medians with interquartile ranges (IQRs). Categorical variables are reported as frequencies (%). The Mann–Whitney U test was used for comparisons of continuous variables, and the The Pearson chi-squared test or Fisher’s exact test was used to evaluate the associations between categorical variables across different groups. This was followed by a Tukey post hoc test for multiple comparisons. The post hoc power analysis was also performed between the two groups. A multivariable logistic regression model was performed, incorporating variables with *p*-values less than 0.1, to identify independent variables. The Cox regression model was used to analyze the time required to achieve CEF. A *p*-value of less than 0.05 was considered statistically significant. All statistical analyses were conducted using SPSS Statistics version 27.0 (Armonk, NY, USA: IBM Corp).

## 3. Results

The enrollment and exclusion flow chart is illustrated in Figure 1. A total of 443 ELBW infants born between January 2017 and December 2021 were enrolled. However, 56 infants who survived less than 7 days, 17 infants with SIP, 2 infants with major GI tract anomalies (1 with ileal atresia and the other with midgut volvulus with a mesenteric defect), 1 infant transferred to another hospital, and 37 infants who did not establish enteral feeding within the first 14 days were excluded. The remaining 330 premature infants were divided into two groups: 206 ELBW infants who received probiotics (probiotic group) and 124 infants who did not (non-probiotic group). Within the probiotic group, 60 infants began receiving probiotics within 14 days of age (subgrouped as early probiotic group), while 146 infants started receiving probiotics after 14 days of age (subgrouped as late probiotic group).

### 3.1. The Characteristic Demographic Data and Comparison of Outcomes between Probiotic Group and Non-Probiotic Group

The demographic data and neonatal characteristics are presented in Table 1. There were no significant differences between the probiotic group and non-probiotic group in terms of GA, BBW, SGA, 1 min and 5 min Apgar scores, EOS, and HsPDA. However, fewer ELBW infants in the probiotic group were fed exclusively human milk (eHM) [18/206 (8.7%) vs. 29/124(23.4%), *p* < 0.01]. Moreover, fewer infants in the probiotic group underwent PDA ligations [32% vs. 43.5%, *p* = 0.03]. Table 2 presents a comparison of outcomes between the probiotic group and non-probiotic group. No significant differences were observed in moderate or severe BPD, NEC, LOS, PVL, or ROP. However, the overall mortality rate was notably lower in the probiotic group compared to the non-probiotic group (4.4% vs. 19.4%, *p* < 0.01). In the multivariable logistic regression model shown in Table 3, after adjusting for eHM and ligation for HsPDA, the odds ratio (OR) for overall mortality in the probiotic group was 0.22 [95% CI 0.09, 0.48] when compared to the non-probiotic group. Regarding deaths attributed to NEC or LOS, the mortality rate was also significantly lower in the probiotic group (1.4% vs. 9.7%, *p* < 0.01) than in the non-probiotic group, and the adjusted OR was 0.12 [95% CI 0.03, 0.45]. Conversely, although the mortality rate not attributed to NEC or LOS was also lower in the probiotic group (2.9% vs. 9.7%, *p* < 0.01) than in the non-probiotic group, there was no significant difference between the two groups after adjustment for eHM and ligation for HsPDA (adjusted OR 0.41, 95% CI 0.84, 7.1). 

### 3.2. The Subgroup Analysis of the Effects on the Timing of Probiotic Initiation

When examining the effects of the timing of probiotic initiation, the neonatal characteristics among the early probiotic group, late probiotic group, and non-probiotic group were compared and are summarized in Table 4. There were no significant differences in GA, BBW, 1 min Apgar score, and 5 min Apgar score. Similarly, a higher proportion of ELBW infants in the non-probiotic group were fed eHM compared to those in the early and late probiotic groups. However, ELBW infants in the early probiotic group tended to initiate feeding earlier and had a lower proportion of HsPDA and ligation for HsPDA. Preterm-related outcomes were compared and are presented in Table 5. No significant differences were observed in moderate BPD, NEC, LOS, PVL and ROP among the three groups. Interestingly, both ELBW infants in the early and late probiotic groups had a significantly lower mortality rate compared to those in the non-probiotic group (0% vs. 6.2% vs. 19.4%, *p* < 0.01). Regarding the causes of mortality, both the early and late probiotic groups showed a significant decrease in the mortality rate attributed to NEC or LOS when compared to the non-probiotic group. However, a significant reduction in mortality rate not attributed to NEC or LOS was only observed in the early probiotic group, not in the late probiotic group, when compared to the non-probiotic group. Additionally, ELBW infants in the early probiotic group had a significantly lower overall mortality rate compared to those in the late probiotic group.

### 3.3. Assess the Effect of Probiotics on the Time to Achieve CEF

In assessing the impact of probiotics on the time to achieve CEF, we used a Cox regression model, taking into account potential influences such as BBW, eHM, ligation for HsPDA, 1 min Apgar score, and mortality rates. As shown in Figure 2, ELBW infants in the early probiotic group had a higher rate of achieving CEF compared to those in the late probiotic and non-probiotic groups. The adjusted odds ratio (OR) for ELBW infants in the early group was 0.68 [95% CI 0.49, 0.94] when compared to the non-probiotic group and 0.56 [95% CI 0.4, 0.76] when compared to the late probiotic group. However, no significant difference was observed between the late probiotic group and the non-probiotic group (*p* = 0.16).

## 4. Discussion

In our study, we found that even in the absence of guidelines and with decisions to administer probiotics primarily at the discretion of clinicians, probiotics continue to provide benefits to preterm infants. Morgan RL et al. reported that combinations of one or more *Lactobacillus* spp. and one or more *Bifidobacterium* spp. could reduce all-cause mortality compared with a placebo (OR, 0.56; 95% CI, 0.39–0.80; high certainty) [6]. Similarly, Yuting Wang et al. also reported the same benefit of the multiple-strain probiotics compared with placebo (risk ratio [RR], 0.69; 95% CI, 0.56 to 0.86) [13]. Our findings support that the primary benefit of probiotics in clinical practice, in reducing mortality rates among preterm infants, is consistent with previous studies. Nevertheless, this benefit is affected by the timing of probiotic administration. Early administration of probiotics not only reduces deaths related to sepsis or NEC, but also decreases mortality from other causes. However, this latter effect in reducing mortality from other cause may not be observed if probiotics are administered late (>14 days old) to preterm infants. Specifically, the most pronounced benefits were observed in preventing deaths related to NEC or LOS. Notably, our study shows that although ELBW infants in the probiotic groups experienced a higher incidence of NEC or LOS, they showed a lower mortality rate. 

Another significant benefit of probiotics is the earlier achievement and higher rate of achieving full enteral feeding (CEF) in our study. In previous studies, the immature development of the intestine in preterm infants has been linked to feeding intolerance, characterized by symptoms such as abdominal distention, gastric residues, or vomiting. This condition not only affects the digestion of nutrients but also contributes to gastrointestinal inflammation, including gastritis, enteritis, and colitis [14,15,16]. Supplementing probiotics can promote the colonization of beneficial microorganisms and fungi, thereby enhancing nutrient absorption and improving gut development. In our study, we found that the beneficial effects of probiotics were only evident in the group that received probiotics early. This could be due to the timing of probiotic administration influencing the establishment of specific probiotic bacteria in the gut. In line with previous findings, starting probiotic supplementation within the first five days after birth has been shown to enhance the detection of beneficial microbiota in preterm infants [17]. This early establishment of beneficial microbiomes may lead to a decrease in pathogenic species and improvements in gut immunity and function [18]. Different species of probiotics have distinct functions in modifying the intestinal immune response. For instance, *Lactobacillus acidophilus* can restrict pathogen colonization and modulate antibodies such as IgA and IgG against pathogenic species or toxins. *Lactobacillus rhamnosus GG* not only protects epithelial cells and keratinocytes from pathogens through competitive exclusion but also secretes antimicrobial substances [19,20]. *Bifidobacterium bifidum* activates toll-like receptor-2 (TLR2) in the intestinal epithelium and enhances cyclooxygenase-2 (COX-2) expression, leading to increased production of prostaglandin E2 (PGE2) in the ileum, which helps protect against intestinal apoptosis associated with NEC [21]. One of the major mechanisms of NEC is the overactivation of TLR-4 in intestinal epithelial cells (IECs) and the subsequent activation of nuclear factor kappa-light-chain-enhancer of activated B cells (NF-κB), resulting in vasoconstriction and intestinal ischemia [22]. In addition to TLR4, IECs also express TLR2, which also signals through the NF-κB pathway [23]. Although the role of TLR2 in NEC is still uncertain, studies have reported that TLR2 might play a protective role in preventing NEC through its modulation of immune responses and interactions with the microbiota in animal models [24].

Due to the synergistic effects of multiple strains, probiotics are believed to modulate immune response and reduce not only overall mortality but also mortality related to NEC and sepsis [25]. In our hospital, mortality not related to NEC or sepsis is predominantly attributed to BPD, whether in combination with pulmonary hypertension or not. Recent studies hypothesize that abnormal inflammatory responses induced by gut microbiota, known as the mucosal response theory, could disrupt lung microbiota and play a role in the pathogenesis of BPD. Previous research has indicated that probiotics may mitigate the risk of BPD in preterm infants [26,27,28]. In our study, ELBW infants in the early probiotic group exhibited the lowest rates of moderate or severe BPD, although this difference was not statistically significant. Conversely, the rate of BPD in the late probiotic group was similar to that in the non-probiotic group. This observation suggests that the effect of probiotics on reducing mortality not associated with NEC or sepsis may only be evident in the early probiotic administration group. In a previous Cochrane review, probiotics were found to significantly reduce the incidence of NEC and sepsis more effectively in preterm infants weighing > 1000 g [9]. However, we did not observe these effects in our study. This discrepancy could potentially be attributed to the variability in the timing of probiotic administration, which ranged from initiation shortly after birth to unspecified times.

Furthermore, our study showed that the clinical use of probiotics in preterm infants, where administration is determined by clinician judgment, can also reduce mortality rates. Early supplementation appears to yield more beneficial effects. However, we did not observe significant effects in preventing NEC, LOS, and BPD. This could be due to the heterogeneous use of probiotics in our study cohort or the sample size of each group not being large enough to identify the significant differences for these morbidities. A recent study showed that gut microbiota is associated multiple organs, which is called the gut–organ axis. Gut dysbiosis in preterm infants may cause several acute morbidities, such as NEC, LOS, BPD, and ROP, and it may also influence long-term outcomes including neurodevelopment and somatic growth [29]. Probiotics, as an intervention affecting the gut microbiota, may serve as a preventive measure for multiple diseases according to this pathogenesis. This underscores the need to establish guidelines for probiotic administration among clinicians. According to the American Academy of Pediatrics, probiotic supplementation may be considered after stable initiation of enteral feeding [30], Similarly, the European Society for Paediatric Gastroenterology Hepatology and Nutrition Committee also provides a conditional recommendation for using selected probiotic strains, such as *Lactobacillus rhamnosus* GG ATCC53103 or a combination of *Bifidobacterium infantis* Bb-02, *Bifidobacterium lactis* Bb-12, and *Streptococcus thermophilus* TH-4, to reduce the risk of NEC [31]. It is imperative for our neonatologists to establish standardized guidelines to optimize probiotic effectiveness and enhance outcomes for preterm infants.

There are several limitations in our study. First, it is a retrospective observational study, which means our findings only support an association between probiotic use and reduced mortality. The different treatment strategies employed for ELBW infants across different time periods, especially in terms of probiotic administration, may have influenced outcomes. Although there were no significant differences in ventilator strategy, enteral feeding consensus, and parental nutrition advice over the past five years, these factors could still have impacted our results. Second, there is a possibility of selection bias due to the initiation of probiotic supplementation based on initial clinical conditions. Third, during our study period, we lacked data on the intestinal microbiome, preventing us from understanding the precise interaction between the gut microbiome and probiotics. The potential of gut microbiota to affect multiple organs and influence acute morbidities via the gut–organ axis in preterm infants has gained increasing attention [29]. Currently, probiotics are the only available intervention for premature infants to modulate gut microbiota. We suggest that the impacts of administering probiotics should be carefully examined in studies about gut microbiota in preterm infants, especially regarding the links with acute morbidities and long-term outcomes. In particular, examining the different effects of various probiotic combinations could pave the way for personalized precision treatment in the future. 

## 5. Conclusions

In conclusion, the administration of probiotics to preterm infants can potentially reduce mortality rates and improve feeding tolerance. Early initiation of probiotics seems to yield greater benefits. Further studies are crucial to clarify the selection of probiotics, the timing of introduction, and the duration of use. Establishing standardized guidelines is also essential for improving outcomes for preterm infants.

## Figures and Tables

**Figure 1 nutrients-16-02995-f001:**
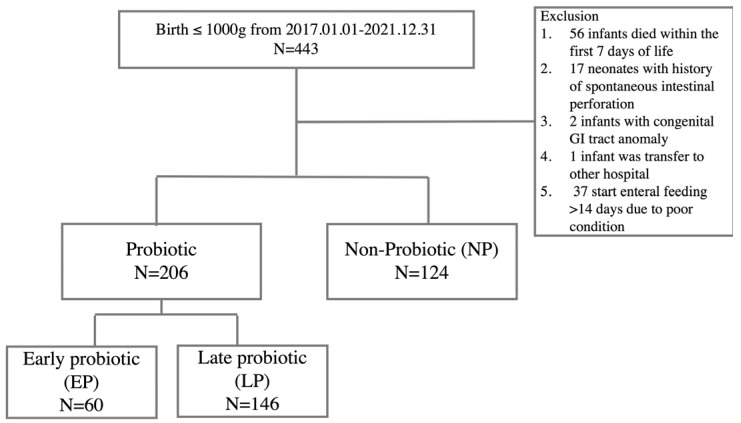
The flow chart of the study.

**Figure 2 nutrients-16-02995-f002:**
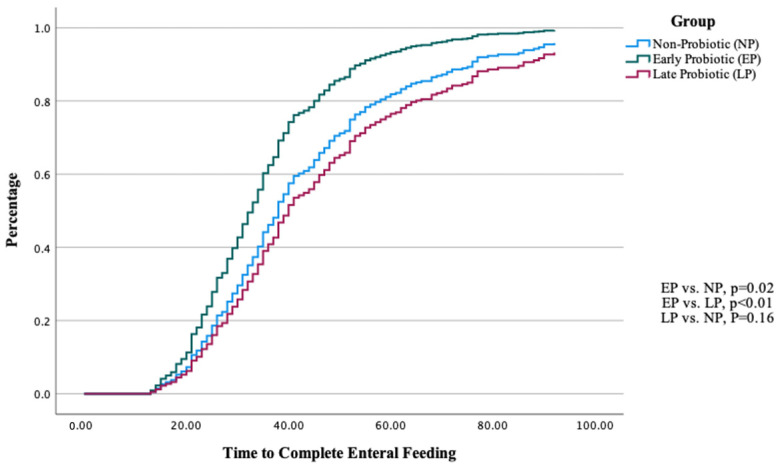
The time–percentage curve illustrates the progression of complete enteral feeding among three groups. The *p*-values were adjusted for birth body weight (BBW), exclusive human milk (eHM) feeding, ligation for hemodynamically significant patent ductus arteriosus (HsPDA), and 1 min Apgar score.

**Table 1 nutrients-16-02995-t001:** The neonatal characteristics between ELBW infants in the probiotic group and those in the non-probiotic group.

Characteristic	Probiotic N = 206	Non-Probiotic (N = 124)	*p*
GA (wk) *	26.7 [25.3, 28.2]	26.3 [25.1, 28.4]	0.30
BBW(gm) *	822 [697, 918]	811 [678, 900]	0.18
SGA	42 (20.4)	29 (23.4)	0.50
1′ Apgar score *	6 [5, 7]	6 [5, 7]	0.31
5′ Apgar score *	8 [7, 9]	8 [7, 9]	0.52
IVH ≥ grade 2	23 (11.2)	16 (12.9)	0.62
eHM	18 (8.7)	29 (23.4)	<0.01
Start feeding age (Day) *	5 [3, 8]	6 [3, 11]	0.10
Early-onset sepsis *	5 (2.4)	3 (2.4)	1.00
HsPDA	97 (47.1)	67 (54.0)	0.20
Receiving medications for HsPDA	31 (15.0)	13(10.3)	0.22
Ligation for HsPDA	66 (32.0)	54 (43.5)	0.03
Receiving medications and ligation for HsPDA	25 (12.0)	20 (15.9)	0.33
Only ligation for HsPDA	41 (19.9)	34 (27.0)	0.13

* Median [IQR], SGA, IVH ≥ grade 2, eHM, EOS, HsPDA, Receiving medications for HsPDA, Ligation for HsPDA, Receving medications and ligation for HsPDA, Only ligation for HsPDA = %. BBW: birth body weight; GA: gestational age; eHM: exclusively human milk; HsPDA: hemodynamically significant patent ductus arteriosus; IVH: intraventricular hemorrhage; SGA: small for gestational age.

**Table 2 nutrients-16-02995-t002:** The preterm outcomes between ELBW infants in the probiotic group and those in the non-probiotic group.

Outcomes	Probiotic N = 206	Non-Probiotic N = 124	*p*	Power
BPD ≥ grade II	136/204 (66.7)	72/102 (70.6)	0.49	0.10
NEC ≥ stage II	17 (8.2)	6 (4.8)	0.65	0.20
Late-onset sepsis	82 (39.8)	40 (32.3)	0.19	0.27
PVL	13 (6.3)	7 (5.6)	0.82	0.04
ROP	38/200 (19)	17/101 (16.8)	0.64	0.07
Mortality, overall	9 (4.4)	24 (19.4)	<0.01	0.99
NEC- or sepsis-related	3 (1.4)	12 (9.7)	<0.01	0.91
Not NEC- or sepsis-related	6 (2.9)	12 (9.7)	0.02	0.73

BPD ≥ grade II; NEC ≥ grade II; LOS; PVL; ROP; Mortality, overall; NEC- or sepsis-related; Not NEC- or sepsis-related = %. BPD: bronchopulmonary dysplasia; NEC: necrotizing enterocolitis; PVL: periventricular leukomalacia; ROP: retinopathy of prematurity.

**Table 3 nutrients-16-02995-t003:** Multivariable logistic regression model assessing mortality risk among ELBW infants in the probiotic group compared to the non-probiotic group.

	OR	95% CI	*p*	aOR ^a^	95% CI	*p*
Mortality, overall	0.19	0.09–0.43	<0.01	0.22	0.09–0.48	<0.01
NEC- or sepsis-related	0.14	0.04–0.50	<0.01	0.12	0.03–0.45	<0.01
Not NEC- or sepsis-related	0.31	0.11–0.86	0.02	0.41	0.84–7.10	0.1

OR: odds ratio; aOR: adjusted odds ratio; CI: confidence interval; ^a^: Adjusted with eHM and ligation for HsPDA.

**Table 4 nutrients-16-02995-t004:** The demographic data between ELBW infants in the early probiotic, late probiotic and non-probiotic groups.

	Early Probiotic (EP), N = 60	Late Probiotic (LP), N = 146	Non-Probiotic (NP), N = 124	*p*-Value
GA (wk) *	27.0 [25.8, 28.5]	26.7 [25.1, 27.9]	26.3 [25.1, 28.4]	0.15
BBW(gm) *	847 [726, 947]	810 [686, 910]	811 [678, 900]	0.06
Age at initiation of probiotics (Day) *	9 [7, 11]	29 [19, 46]	-	<0.01 ^a^
SGA	13 (21.7)	19 (19.9)	29 (23.4)	0.78
1′ Apgar score *	7 [6, 7]	6 [5, 7]	6 [5, 7]	0.06
5′ Apgar score *	8 [8, 9]	8 [7, 9]	8 [7, 9]	0.19
IVH ≥ grade 2	5 (8.3)	18 (12.3)	16 (12.9)	0.64
eHM	6 (10)	12 (8.2)	29 (23.4)	<0.01 ^a,b^
Start feeding age (Day) *	4 [3, 6]	6 [4, 9]	6 [3, 11]	<0.01 ^a,c^
Early-onset sepsis	0 (0)	5 (3.4)	3 (2.4)	0.35
HsPDA	18 (30.0)	79 (54.1)	67 (54.0)	<0.01 ^a,c^
Receiving medications for HsPDA	7 (11.7)	24 (16.4)	13 (10.3)	0.32
Ligation for HsPDA	11 (18.3)	55 (37.7)	54 (43.5)	<0.01 ^a,c^
Receiving medications and ligation for HsPDA	2 (3.3)	23 (15.8)	20 (15.9)	<0.01 ^a,c^
Only ligation for HsPDA	9 (15.0)	32 (21.9)	34 (27.0)	0.16

* Median [IQR], SGA, IVH ≥ grade 2, eHM, EOS, HsPDA, Receiving medications for HsPDA, Ligation for HsPDA, Receving medications and ligation for HsPDA, Only ligation for HsPDA = %. a: significant difference between EP and NP. b: significant difference between LP and NP. c: significant difference between EP and LP. GA: gestational age; BBW: birth body weight; eHM: exclusively human milk; HsPDA: hemodynamically significant patent ductus arteriosus; IVH: intraventricular hemorrhage; SGA: small for gestational age.

**Table 5 nutrients-16-02995-t005:** The outcomes between ELBW infants in the early probiotic, late probiotic and non-probiotic groups.

	Early Probiotic (EP), N = 60	Late Probiotic (LP), N = 146	Non-Probiotic (NP), N = 124	*p*-Value
BPD ≥ grade II (%)	35 (58.3)	101/144 (70.1)	72/102 (70.6)	0.20
NEC ≥ stage II (%)	23 (38.3)	59 (40.4)	40 (32.3)	0.37
Late-onset sepsis (%)	4 (6.7)	9 (6.2)	7 (5.6)	0.96
PVL (%)	4 (6.7)	13 (8.9)	6 (4.8)	0.42
ROP (%)	8 (13.3)	30/140 (21.4)	17/101 (16.8)	0.35
Mortality, overall (%)	0 (0)	9 (6.2)	24 (19.4)	<0.01 ^a,b,c^
NEC- or sepsis-related (%)	0 (0)	3 (1.4)	12 (9.7)	<0.01 ^a,b^
Not NEC- or sepsis-related (%)	0 (0)	6 (4.1)	11 (8.4)	0.03 ^a^

BPD ≥ grade II, NEC ≥ grade II, LOS, PVL, ROP, mortality, overall, NEC- or sepsis-related, Not NEC-or sepsis-related = %. a: significant difference between EP and NP. b: significant difference between LP and NP. c: significant difference between EP and LP. BPD: bronchopulmonary dysplasia; NEC: necrotizing enterocolitis; PVL: periventricular leukomalacia; ROP: retinopathy of prematurity.

## Data Availability

Data may be available upon request to the corresponding author because of legal reasons.

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
