# Peer review of "Impact of Clinical Use of Probiotics on Preterm-Related Outcomes in Infants with Extremely Low Birth Weight"

_nutrients, 2024, doi:10.3390/nu16172995_

Round 1

Reviewer 1 Report

Comments and Suggestions for Authors

In this retrospective study, the authors examine the effect of administering probiotics in preterm infants to reduce mortality. The planning of the study, the statistical analysis, the presenting of the results in tables and figures, the list of references are good. The clinical effect of probiotics is pointed out very well. The English language is good. The limitations of the study is pointed out. It should be added, in what studies, the effects of administering probiotics might be better examined.

Comments on the Quality of English Language

The English language is good and qualified.

Author Response

Comments 1:

In this retrospective study, the authors examine the effect of administering probiotics in preterm infants to reduce mortality. The planning of the study, the statistical analysis, the presenting of the results in tables and figures, the list of references are good. The clinical effect of probiotics is pointed out very well. The English language is good. The limitations of the study is pointed out. It should be added, in what studies, the effects of administering probiotics might be better examined.

Response 1:

Thank you for your encouragement and valuable opinion. Recent studies have shown that the gut microbiota is associated with multiple organs, forming what is known as the gut-organ axis. Gut dysbiosis in preterm infants may cause several acute morbidities. Nevertheless, as an intervention affecting the gut microbiota, probiotics serve as both a preventive measure and a treatment for multiple diseases by addressing this pathogenesis. The microbiome should be examined through controlled studies. Additionally, longitudinal studies that follow the long-term outcomes of preterm infants receiving probiotics could offer valuable insights into the sustained benefits or potential risks associated with their use.

We revised our manuscript in Chapter 4. Discussion, row 315-322” These limitations highlight the need for further prospective studies with controlled designs and comprehensive gut microbiome analyses to better elucidate the effects of probiotics in premature infants. Additionally, longitudinal studies that follow the long-term outcomes of preterm infants receiving probiotics could offer valuable insights into the sustained benefits or potential risks associated with their use.”

Reviewer 2 Report

Comments and Suggestions for Authors

Many studies and meta-analyses on probiotics in the population of ELBWI have been published to date, however the results are still not unanimously positive. Therefore, I find the paper by Wu W-H et al very useful, especially from the point of view of clinical applicability. I have no major comments or wishes for changes regarding the content of the contribution, but some additional information would be useful.

The first question is related to probiotic supplements mentioned and their administration in practise - as I find the statement that "The choice of regimen and the timing of introduction are determined by the responsible neonatologists" in lines 77-78 insufficiently determined and could even mean a bias to the results due to the substantial heterogeneity of administration. More specifically: what was the individual dose (whole 150 mg of MoProbi-LR or 250 mg of Infloran or influenced by infant's weight?), was it given in a single or divided dose, diluted with milk or aqua, given orally or by gastric tube?

If data on exact timing of probiotic introduction exist (day of life, particulrly in the early group), indicating the average age (and range) at probiotic initiation would be useful for easier following of research results; again I comment: description < 2 weeks for early and > 2 weeks for late probiotics is insufficiently precise.

Some specific comments:

·      more precise definition of EOS - LOS is needed (was positive blood or CSF culture obligatiory for the diagnosis?)

·      as the incidence of HsPDA that needed ligation is extremely high, I wonder about the attempt(s) to close PDA with medications (ibuprofen, indomethacin, paracetamol) before decision for surgical ligation

·      unification of titles: eg in Figure 1: Probiotic instead of With Probiotics, in Tables 1 / 2 Non-Probiotic instead of Non-probiotic

·      in all Tables I miss description eg. at GA and BBW = mean / range, and SGA, IVH, eHM, EOS, ...  = %

·      in Tables 4 and 5 the column Non-Probiotic is redundant, as the data have been already presented previously and have not been included in statistical analysis in theseTables - only comparison of EP and LP; if removed, the titles of both Tables should be changed

·      in Conflicts of Interest: the letter A at the beginning and double final dot are redundant (typos).

I also wonder if the statements Institutional Review Board Statement ("requirement for informed consent was waived")  and Informed Consent Statement ("informed consent was obtained from all subjects...") are not contradictory.

Congratulations for up-to-date and relevant citations.

Author Response

Comments and Suggestions for Authors

Many studies and meta-analyses on probiotics in the population of ELBWI have been published to date, however the results are still not unanimously positive. Therefore, I find the paper by Wu W-H et al very useful, especially from the point of view of clinical applicability. I have no major comments or wishes for changes regarding the content of the contribution, but some additional information would be useful.

Comments 1:

The first question is related to probiotic supplements mentioned and their administration in practise - as I find the statement that "The choice of regimen and the timing of introduction are determined by the responsible neonatologists" in lines 77-78 insufficiently determined and could even mean a bias to the results due to the substantial heterogeneity of administration. More specifically: what was the individual dose (whole 150 mg of MoProbi-LR or 250 mg of Infloran or influenced by infant's weight?), was it given in a single or divided dose, diluted with milk or aqua, given orally or by gastric tube?

Response 1:

Thank you for pointing this out. In our hospital, we typically administer a single dose of Infloran 250 mg (250 mg/capsule, containing Lactobacillus acidophilus 150 × 10⁹ cfu/g and Bifidobacterium bifidum 150 × 10⁹ cfu/g) or MoProbi-LR (150 mg/capsule, containing Lactobacillus rhamnosus GG 10⁹ cfu/g) once a day, diluted with milk and delivered via an oral-gastric tube. Consequently, we have added the following sentence to Chapter 2. Material and Methods row 74-48:

The choice of probiotics regimen is MoProbi-LR (150mg/cap, containing Lactobacillus rhamnosus GG 109 cfu/g) or Infloran (250mg/cap, containing Lactobacillus acidophilus 150x109 cfu/g and Bifidobacterium bifidum 150x109 cfu/g) for a single dose, which is typically administered once a day via oral gastric tube diluted with milk”

If data on exact timing of probiotic introduction exist (day of life, particulrly in the early group), indicating the average age (and range) at probiotic initiation would be useful for easier following of research results; again I comment: description < 2 weeks for early and > 2 weeks for late probiotics is insufficiently precise.

Response 2:

Thank you for your valuable input. In our cohort, the age at which probiotic was initiated in the early group had a median of 9 days old, with an interquartile range (IQR) of [7, 11] days. In the late group, probiotic was initiated at the median age of 29 days with IQR [19,46]. We have added these results to Table 4.

Early probiotic
(EP), N=60

Late probiotic
(LP), N=146

Non-Probiotic
(NP), N=124

P-value

GA (wk)*

27.0 [25.8, 28.5]

26.7[25.1,27.9]

26.3 [25.1, 28.4]

0.15

BBW(gm)*

847 [726, 947]

810[686,910]

811 [678, 900]

0.06

Age at initiation of probiotics (Day)*

9 [7,11]

29 [19,46]

-

<0.01a

SGA

13 (21.7)

19 (19.9)

29 (23.4)

0.78

1’ Apgar score*

7 [6,7]

6 [5,7]

6 [5,7]

0.06

5’ Apgar score*

8 [8,9]

8 [7,9]

8 [7,9]

0.19

IVH ≧ grade 2

5 (8.3)

18 (12.3)

16 (12.9)

0.64

eHM

6 (10)

12 (8.2)

29 (23.4)

<0.01a,b

Start feeding age (Day)*

4 [3,6]

6 [4, 9]

6 [3, 11]

<0.01a,c

Early-onset sepsis (%)

0 (0)

5 (3.4)

3 (2.4)

0.35

HsPDA

18 (30.0)

79 (54.1)

67 (54.0)

<0.01a,c

Receiving medications for HsPDA

7 (11.7)

24 (16.4)

13(10.3)

0.32

Ligation for HsPDA

11 (18.3)

55 (37.7)

54 (43.5)

<0.01a,c

Receiving medications and ligation for HsPDA

2 (3.3)

23 (15.8)

20(15.9)

<0.01a,c

Only Ligation for HsPDA

9 (15)

32 (21.9)

34 (27.0)

0.16

Table 4

Some specific comments:

Comment3:    more precise definition of EOS - LOS is needed (was positive blood or CSF culture obligatiory for th e diagnosis?)

Response 3:

Thank you for pointing this out. We used the definition from AAP that the EOS and LOS were determined based on the timing of onset and microbiological results from blood or CSF, with EOS developing within the first 3 days of life and LOS occurring after 3 days of life. We retrospectively reviewed our manuscript that the definition we provided was a typo. The correct sentence was added to Chapter 2. Material and Methods, row 88-90
EOS and LOS were determined based on the timing of onset and microbiological results from blood or CSF, with EOS developing within the first 3 days of life and LOS occurring after 3 days of life.”

Comment 4.      as the incidence of HsPDA that needed ligation is extremely high, I wonder about the attempt(s) to close PDA with medications (ibuprofen, indomethacin, paracetamol) before decision for surgical ligation

Response 4:

Thank you for your valuable opinion. In our cohort, 25 cases in the Probiotic group and 20 cases in the non-Probiotic group received medications before surgical ligation. In the subgroup analysis, 2 cases in the early Probiotic group and 23 cases in the late Probiotic group received medications before surgical ligation. There were no differences between each group when compared using the chi-square test. We have revised the demographic data of PDA treatment in Table 1 and Table 4.

In our institution, the medications for PDA treatment were limited. Only oral acetaminophen, oral ibuprofen, and IV ibuprofen were available. IV Propacetamol or paracetamol, which are more suitable for ELBW infants with acute kidney injury who cannot tolerate feeding, were not available in our institution until this year. As a result, the surgical ligation rate was relatively high in the past.

Characteristic

Probiotic
N=206

Non-Probiotic
(N=124)

P

GA (wk)*

26.7 [25.3, 28.2]

26.3 [25.1, 28.4]

0.30

BBW(gm)*

822 [697, 918]

811 [678, 900]

0.18

SGA

42 (20.4)

29 (23.4)

0.50

1’ Apgar score*

6 [5,7]

6 [5,7]

0.31

5’ Apgar score*

8 [7,9]

8 [7,9]

0.52

IVH ≧ grade 2

23 (11.2)

16 (12.9)

0.62

eHM

18 (8.7)

29 (23.4)

<0.01

Start feeding age (Day)*

5 [3, 8]

6 [3, 11]

0.10

Early-onset sepsis*

5 (2.4)

3 (2.4)

1.00

HsPDA

97 (47.1)

67 (54.0)

0.20

Receiving medications for HsPDA

31 (15.0)

13 (10.3)

0.22

Ligation for HsPDA

66 (32.0)

54 (43.5)

0.03

Receiving medications and ligation for HsPDA

25 (12.0)

20 (15.9)

0.33

Only Ligation for HsPDA

41 (19.9)

34 (27.0)

0.13

Table1.

Table4.

Early probiotic
(EP), N=60

Late probiotic
(LP), N=146

Non-Probiotic
(NP), N=124

P-value

GA (wk)*

27.0 [25.8, 28.5]

26.7[25.1,27.9]

26.3 [25.1, 28.4]

0.15

BBW(gm)*

847 [726, 947]

810[686,910]

811 [678, 900]

0.06

Age at initiation of probiotics (Day)*

9 [7,11]

29 [19,46]

-

<0.01a

SGA

13 (21.7)

19 (19.9)

29 (23.4)

0.78

1’ Apgar score*

7 [6,7]

6 [5,7]

6 [5,7]

0.06

5’ Apgar score*

8 [8,9]

8 [7,9]

8 [7,9]

0.19

IVH ≧ grade 2

5 (8.3)

18 (12.3)

16 (12.9)

0.64

eHM

6 (10)

12 (8.2)

29 (23.4)

<0.01a,b

Start feeding age (Day)*

4 [3,6]

6 [4, 9]

6 [3, 11]

<0.01a,c

Early-onset sepsis (%)

0 (0)

5 (3.4)

3 (2.4)

0.35

HsPDA

18 (30.0)

79 (54.1)

67 (54.0)

<0.01a,c

Receiving medications for HsPDA

7 (11.7)

24 (16.4)

13(10.3)

0.32

Ligation for HsPDA

11 (18.3)

55 (37.7)

54 (43.5)

<0.01a,c

Receiving medications and ligation for HsPDA

2 (3.3)

23 (15.8)

20(15.9)

<0.01a,c

Only Ligation for HsPDA

9 (15)

32 (21.9)

34 (27.0)

0.16

Comment 5. unification of titles: eg in Figure 1: Probiotic instead of With Probiotics, in Tables 1 / 2 Non-Probiotic instead of Non-probiotic.  in all Tables I miss description eg. at GA and BBW = mean / range, and SGA, IVH, eHM, EOS, ...  = %

Response 5:

Thank you for pointing this out. We sincerely appreciate your careful review. We unified titles in Figure1 and Table 1 and 2 and added the unit in the tables in resubmitted manuscripts.

Comment 6:     in Tables 4 and 5 the column Non-Probiotic is redundant, as the data have been already presented previously and have not been included in statistical analysis in theseTables - only comparison of EP and LP; if removed, the titles of both Tables should be changed

Response 6:

Thank you for your valuable advice. In our study, we aimed to investigate how the timing of probiotic initiation affects its efficacy. We used the non-Probiotic group as a control group to compare the benefits associated with different initiation timings. To perform these comparisons, we employed the Tukey post hoc test for multiple comparisons. Specifically, we compared the differences among the Early Probiotic vs. non-Probiotic group, the Late Probiotic vs. non-Probiotic group, and the Early Probiotic vs. Late Probiotic group. This analysis allowed us to better understand the potential impact of timing on the efficacy of probiotic administration.  

Comment 7.      in Conflicts of Interest: the letter A at the beginning and double final dot are redundant (typos).

Thank you for pointing this out. We sincerely appreciate your careful review. We had deleted the letter at the beginning and the double final dot. As a result the corrected conflicts of interests in our resubmitted manuscripts is over row 304-305 “The authors affirm that there are no conflicts of interest associated with this research.”

Comment 8. I also wonder if the statements Institutional Review Board Statement ("requirement for informed consent was waived")  and Informed Consent Statement ("informed consent was obtained from all subjects...") are not contradictory.

Response 8: Thank you for pointing this out. We sincerely appreciate your careful review. There was a formatting mistake in previous manuscript when using the template. As a result, the informed consent statement was deleted in our resubmitted manuscript.

Congratulations for up-to-date and relevant citations.

Reviewer 3 Report

Comments and Suggestions for Authors

Summary statement

Running title: Impact of Clinical Use of Probiotics on Preterm-Related Outcomes in Infants with Extremely Low Birth Weight

The manuscript aims to investigate whether premature infants still derive benefits from probiotic supplementation based on individual clinician judgment in the absence of consistent guidelines. It also explores whether the timing of probiotic initiation impacts its efficacy.

This article's main and innovative finding is the investigation of the benefits of probiotic use for preterm infants based on physician judgment. And also examining the time at which the use of probiotics begins concerning their efficacy in premature babies.

The manuscript is relevant, and the objective is clear, it also has an interesting and original approach, but we suggest that the authors review a few points:

  1. Abstract

Check according to the journal's rules, the abstract should be a total of about 200 words maximum.

  1. Introduction

It is a great challenge for practices and guidelines to focus on such a specific niche, so your study is interesting and valid, many societies and countries present guidelines that can mirror ideas for your hospital and some systematic reviews present a robust body of evidence.

Are the authors interested in creating a guideline for their hospital? It would be very worthwhile and publishing it would open the door to new ideas for guidelines for specific hospitals in specific niches that may be related to regionalization, and level of development, among others…

  1. Materials and Methods

General: What was the sample size calculation for the study? It is important to report it.

Authors should include a nutritional table for the two probiotic formulations used in this study.

  1. Results

Figure 1. The flow chart of the study: “56 infants expired ≤ 7 days of age”, although the sentence is concise and communicates the information, it is an informal sentence for an academic context. Suggested phrases: 56 infants died within the first 7 days of life, or 56 infants died ≤ 7 days of age. It is essential to present the data on infant mortality in your figure clearly and concisely, avoiding terms that give an informal or ambiguous tone to your figure.

  1. Discussion

L 202: Authors should include previous studies (main findings) in this topic. For example: "previous studies have shown... and provide more than two studies to corroborate their findings.

L 214-220: Authors should revise these paragraphs to avoid using abbreviations such as PGE2, TLR-2, and COX-2 without actually writing the full terms. The three molecules are closely linked in the inflammatory response and deserve a more elaborate explanation of their role in the context of the topic.

General: Authors should review the formatting of the discussion text.

Author Response

Comments 1: Abstract

Check according to the journal's rules, the abstract should be a total of about 200 words maximum.

Response1:

Thank you for pointing this out. After thorough review and modification, the abstract is now 200 words.

Comments 2: Introduction

It is a great challenge for practices and guidelines to focus on such a specific niche, so your study is interesting and valid, many societies and countries present guidelines that can mirror ideas for your hospital and some systematic reviews present a robust body of evidence.

Are the authors interested in creating a guideline for their hospital? It would be very worthwhile and publishing it would open the door to new ideas for guidelines for specific hospitals in specific niches that may be related to regionalization, and level of development, among others…

Response 2:

Thank you for your encouragement. It is one of our motivations for conducting this retrospective study. We believe this study will provide our staff with valuable information and evidence, helping to build consensus and develop guidelines for the use of probiotics in our neonatal intensive care units.

Materials and Methods

Comments 3:

General: What was the sample size calculation for the study? It is important to report it.

Authors should include a nutritional table for the two probiotic formulations used in this study.

Response 3:

Thank you for pointing this out. We retrospectively reviewed the cohort to compare the timing of probiotic administration. To minimize the effects of changes in clinical practice, we only conducted a 5-year period review in this study. Because we already enrolled all ELBW infants in this 5-year cohort, we did not calculate the sample size before the study. However, we strongly agree with your opinion that enrolling an adequate sample size is very important to guarantee enough power to differentiate the variables. We calculated the power using the post-hoc power analysis method, and the results are presented in Table 2. The method’s description was also added in chapter 2 row 106-107: “The post-hoc power analysis was also performed between the two groups

Table 2.

Outcomes

Probiotic

N=206

Non-Probiotic
N=124

P

Power

BPD ≥ grade II

136/204 (66.7)

72/102 (70.6)

0.49

0.11

NEC ≥ stage II

17 (8.2)

6 (4.8)

0.65

0.20

Late-onset sepsis

82 (39.8)

40 (32.3)

0.19

0.27

PVL

13 (6.3)

7 (5.6)

0.82

0.04

ROP

38/200 (19)

17/101 (16.8)

0.64

0.07

Mortality, overall

9 (4.4)

24 (19.4)

<0.01*

0.99

NEC or sepsis related

3 (1.4)

12 (9.7)

<0.01*

0.91

Not NEC or sepsis related  

6 (2.9)

12 (9.7)

0.02

0.73

Results

Comments 4:

Figure 1. The flow chart of the study: “56 infants expired ≤ 7 days of age”, although the sentence is concise and communicates the information, it is an informal sentence for an academic context. Suggested phrases: 56 infants died within the first 7 days of life, or 56 infants died ≤ 7 days of age. It is essential to present the data on infant mortality in your figure clearly and concisely, avoiding terms that give an informal or ambiguous tone to your figure.

Response 4.

Thank you for your precious opinion. We change the description in our flow chart of figure 1.

Discussion

Comments 5: L 202: Authors should include previous studies (main findings) in this topic. For example: "previous studies have shown... and provide more than two studies to corroborate their findings.

Response 5:

Thank you for point this out. Both studies are systematic review focusing on the effects of probiotic. Morgan RL et al. reported that combinations of one or more Lactobacillus spp and one or more Bifidobacterium spp could reduce all-cause mortality compared with placebo (OR, 0.56; 95% CI, 0.39–0.80; high certainty). Similarly, Yuting Wang et al. also reported that the same benefit of the multiple-strain probiotics compared with placebo (risk ratio [RR], 0.69; 95% CI, 0.56 to 0.86). Our findings support that the primary benefit of probiotics in clinical practice in reducing mortality rates. Nevertheless, we note that this effect is influenced by the timing of probiotic administration. We have revised the manuscript in Chapter 4, discussion, rows 224-234.Morgan RL et al. reported that combinations of one or more Lactobacillus spp and one or more Bifidobacterium spp could reduce all-cause mortality compared with placebo (OR, 0.56; 95% CI, 0.39–0.80; high certainty).[6] Similarly, Yuting Wang et al. also reported that the same benefit of the multiple-strain probiotics compared with placebo (risk ratio [RR], 0.69; 95% CI, 0.56 to 0.86).[13] Our findings support that the primary benefit of probiotics in clinical practice, in reducing mortality rates among preterm infants, is consistent with previous studies. Nevertheless, this benefit is affected by the timing of probiotic administration. Early administration of probiotics not only reduces deaths related to sepsis or NEC, but also decreases mortality from other causes. However, this latter effect in reducing mortality from other cause may not be observed if probiotics are administered late (>14 days old) to preterm infants.

Comments 6: L 214-220: Authors should revise these paragraphs to avoid using abbreviations such as PGE2, TLR-2, and COX-2 without actually writing the full terms. The three molecules are closely linked in the inflammatory response and deserve a more elaborate explanation of their role in the context of the topic.

Response 6:

Thank you for point this out. We revised the full terms without abbreviations. The revised sentences are in Chapter 4. Discussion, Row 257-261: ”Bifidobacterium bifidum activates toll-like receptor-2 (TLR-2) in the intestinal epithelium and enhances cyclooxygenase-2 (COX-2) expression, leading to increased production of prostaglandin E2 (PGE2) in the ileum, which helps protect against intestinal apoptosis associated with NEC. One of the major mechanisms of NEC is the overactivation of TLR-4 in intestinal epithelial cells (IECs) and the subsequent activation of nuclear factor kappa-light-chain-enhancer of activated B cell (NF-κB), resulting in vasoconstriction and intestinal ischemia.[citation: doi: 10.1016/j.jcmgh.2018.04.001] In addition to TLR4, IECs also express TLR2, which also signals through the NF-κB pathway.[Citation, https://doi.org/10.1016/j.mucimm.2023.02.002] Although the role of TLR2 in NEC is still uncertain, studies have reported that TLR2 might play a protective role in preventing NEC through its modulation of immune responses and interactions with the microbiota in animal models . [ 10.1016/j.jamcollsurg.2006.05.126] “

Comment 7. General: Authors should review the formatting of the discussion text.

Response 7:

Thank you so much. We restructured (sequenced) the discussion.
